# How Sports Involvement and Brand Fit Influence the Effectiveness of Sports Sponsorship from the Perspective of Predictive Coding Theory: An Event-Related Potential (ERP)-Based Study

**DOI:** 10.3390/brainsci14090940

**Published:** 2024-09-20

**Authors:** Haonan Shi, Li Zhang, Hongfei Zhang, Jianlan Ding, Zilong Wang

**Affiliations:** 1School of Sports Economics and Sports Management, Xi’an Physical Education University, Xi’an 710064, China; 15689491213@163.com (H.S.); 106034@tea.xaipe.edu.cn (L.Z.); 17200604317@163.com (Z.W.); 2School of Foreign Language, Chengdu Sport University, Chengdu 610041, China; zhfht@163.com; 3Institute of Sports Neuromanagement and Social Behavioral Decision, Xi’an Physical Education University, Xi’an 710064, China

**Keywords:** sports sponsorship, predictive coding theory, sports involvement, ERP technology

## Abstract

**Background/Objectives:** With the rapid expansion of the global sports market, the significance of sports sponsorship has attracted growing attention. However, during the golden age of the sports industry’s development in China, international sports brand giants such as Nike, Adidas, and Under Armour have rapidly captured a substantial share of the Chinese sports consumer market through their distinctive product designs and varied marketing strategies. This has resulted in a highly competitive environment for China’s sports goods industry. Therefore, fostering the improved development of domestic sports brands has become a crucial issue deserving of thorough scholarly investigation. This study examines how consumers’ differing levels of sports involvement and the degree of fit between the sponsoring brand and the sponsored event affect their cognitive and emotional responses to sports sponsorships. **Methods:** By employing Predictive Coding Theory and ERP (event-related potential) brainwave technology, this study delves into the psychological and neurobiological levels to analyze the impact of consumer sports involvement on the processing of sponsorship information. **Results:** The results indicate significant differences in cognitive and emotional responses between high-involvement and low-involvement consumers. Additionally, the fit between the sponsoring brand and the sponsored event also significantly affects consumers’ cognitive and emotional responses. These differences stem from consumers’ complex and sophisticated predictive coding models. **Conclusions:** This study not only provides scientific evidence for sports brands in selecting and executing sponsorship activities, but also offers new perspectives for evaluating and optimizing sponsorship effectiveness.

## 1. Introduction

In recent years, with the rapid development of the global sports market, sports sponsorship has become an indispensable part of sports enterprises. Many sports brands have increased their investment in sponsoring sports events. In 2020, the global sports sponsorship market was valued at an estimated USD 57 billion, and it is expected to grow to nearly USD 90 billion by 2027. Undoubtedly, sponsoring brands have high expectations for the outcomes of their sponsorships of various sports events. They aim to achieve multiple effects through sponsoring sports events, such as brand exposure, association, target audience attraction, brand value transmission, and a competitive differentiation advantage, to enhance their brand influence and market competitiveness. However, individual biases towards sponsor brands related to events can affect perceptions [1]. Consumers are more likely to recognize a brand as a sponsor and develop positive feelings towards it when there is a clear connection between the brand and the event (e.g., a tennis racket brand sponsoring a tennis match) [2,3]. Conversely, if there is no apparent connection (e.g., an agricultural machinery company sponsoring a tennis match), consumers’ attitudes may be less favorable. For example, in the NFL, Nike currently dominates player sponsorship and holds exclusive rights to provide jerseys and sideline apparel for all 32 NFL teams until 2028. In contrast, Adidas only supports a few individual players. This difference illustrates how the alignment between a brand and an event affects consumer brand perceptions and attitudes. Nike’s extensive sponsorship and long-term agreements enhance its association with NFL events, increasing brand exposure and positive consumer sentiment. Although Adidas’s strategy ensures its market positioning, its limited support scope may not significantly strengthen overall brand recognition and emotional connection within the NFL. This further underscores the crucial role of brand–event alignment in sponsorship effectiveness.

In exploring the effectiveness of sports sponsorship, academia has identified several influencing factors. These factors include brand awareness, the fit between the sponsorship activity and the event, and consumer brand attitudes and loyalty [4]. Among these, the fit between the brand and the event is considered one of the key factors affecting sponsorship effectiveness. Highly matched sponsorship activities can enhance consumers’ brand memory and associations, thereby improving sponsorship effectiveness [5]. Additionally, the quality of sponsorship execution, media exposure, and consumer brand loyalty also influence the effectiveness of sponsorship to varying degrees [6]. To provide a comprehensive analysis, this study examines these factors in the context of their multidimensional impact on sponsorship effectiveness. Here, “sponsorship effectiveness” is defined as the multidimensional market impact a brand achieves through sports sponsorship activities on its target audience. Specifically, it includes the enhancement of brand awareness (how sponsorship increases brand recognition and visibility), audience acceptance (the degree to which the audience associates the brand with the sponsored sports), and changes in consumer attitudes (emotional responses and shifts in behavioral intentions following exposure to sponsorship). Although brand awareness is a key measurement indicator in this study, our research extends beyond this scope. By analyzing brand fit and consumer involvement, and employing event-related potential (ERP) technology alongside subjective ranking data, we explore how these factors impact consumers’ cognitive and emotional responses to brands. These insights provide a comprehensive understanding of sponsorship effectiveness. While not all aspects of sponsorship effectiveness are directly measured, our findings offer valuable theoretical insights for brands in planning and executing sponsorship activities, especially in selecting sports events that align with their brand image and identifying target audience segments.

Despite the importance of these factors, academia has increasingly recognized that consumers’ sports involvement is an even more critical factor. Sports involvement is a concept reflecting an individual’s interest and participation in sports activities. It encompasses not only the individual’s interest and frequency of participation in specific sports activities, but also their emotional connection to these activities, as well as their centrality (the importance of sports in the life of the consumer) and symbolic value (the significance of sports activities to the consumer’s identity). Research shows that highly involved sports consumers usually have a deeper understanding and stronger interest in the sports field. This deep involvement is not only reflected in frequent participation in sports activities, but also in the emotional investment and identification with sports. For example, a fervent football fan may regularly watch matches, actively participate in football-related social activities, collect memorabilia, and consider football a part of their social identity [7,8]. Consequently, they form more positive attitudes and emotional connections toward the sponsoring brand. In contrast, consumers with low sports involvement may have weaker cognitive and emotional responses to the sponsoring brand due to a lack of relevant knowledge and emotional connection, thereby affecting the realization of sponsorship effectiveness [9]. Understanding these differences in sports involvement helps explain why some well-known brands fail to achieve expected results when sponsoring major sports events. Therefore, in-depth research into consumers’ sports involvement and its impact on sponsorship effectiveness is an important direction for optimizing sports sponsorship strategies.

Although existing research has confirmed that different levels of sports involvement and the fit between the brand and the event affect consumers’ emotions and attitudes towards sports events, studies on the interaction between brand fit and sports involvement remain limited [10]. This research gap results in a lack of scientific basis for brands when selecting and executing sponsorship activities, making it difficult to effectively identify and utilize the compatibility between the brand and sports activities to enhance sponsorship effectiveness [11]. Additionally, the lack of in-depth exploration of the relationship between brand fit and sports involvement may lead to an incomplete consideration of these two key factors when evaluating sponsorship effectiveness, thereby affecting the efficient allocation and use of marketing resources [6]. In fact, understanding how the differences in knowledge, experience, and emotions brought by different levels of sports involvement influence consumers’ perceptions of and preferences towards sports products, and how these further affect their purchase motivations and decision-making processes in relation to brands with varying degrees of fit, is crucial for accurately assessing the effectiveness of sponsorship activities by sports sponsorship companies.

In this context, this paper intends to apply perceived fit theory and Predictive Coding Theory to sports brand sponsorship research. The aim is to explore how the interaction between different levels of brand fit and sports involvement affects the effectiveness of sports sponsorship, as well as understanding the underlying neural mechanisms of this influence. The core concept in perceived fit theory is “fit,” which refers to whether consumers subjectively perceive that a particular option matches their preferences, needs, and expectations [12]. A high fit implies a stronger association between the brand and the sports activity, thereby enhancing consumers’ cognitive and emotional connections to the brand [13]. Perceived fit theory explains how the fit between the brand and the sports activity influences consumers’ attitudes and behaviors: when the brand and the sports activity have a high degree of fit, consumers are more likely to transfer their existing positive emotions to the brand, thereby enhancing their cognitive and emotional connections to it. Meanwhile, consumers’ sports involvement affects their perceptions of and reactions to the fit between the brand and the sports activity. Hence, high-involvement consumers are more likely to respond positively to highly matched sponsorship activities.

Predictive Coding Theory was initially used to explain how the brain processes information and predicts external stimuli [14]. It has since been widely applied in marketing and brand sponsorship research. In these fields, the theory is used to understand how consumers process brand information based on past experiences and knowledge, and how this information influences their attitudes and behaviors [15]. For example, consumers with a high level of involvement in basketball may have deeper cognitive and emotional responses to basketball-related sponsorship activities, which may be closely related to their attitudes towards the brand and their purchase intentions [16]. Therefore, the application of Predictive Coding Theory in the field of sports sponsorship can help brands more accurately target their sponsorship activities, ensuring that sponsorship information better matches the sports involvement and expectations of the target consumer group. This matching can enhance consumers’ cognitive and emotional connections to the brand, promoting more effective market communication and consumer engagement [17].

Thus, this study combines these two theories to analyze the cognitive and emotional responses of consumers with different levels of sports involvement when faced with sponsorships from brands with varying degrees of fit. The research then assesses the impact of these responses on sponsorship effectiveness. By employing ERP (event-related potential) technology, we will be able to directly measure consumers’ brain activity when they encounter sports brands and events with various degrees of fit. This method provides a direct and objective measurement of consumers’ cognitive and emotional processing, revealing the neural-level differences in how consumers with varying levels of sports involvement process sponsorship information.

The goal of this paper is to provide sports brands with deeper market insights and strategic guidance, helping them more effectively select and execute sponsorship activities. By understanding how consumers with different levels of sports involvement process sponsorship information at the neural level when faced with varying degrees of brand fit, brands can develop more precise marketing strategies, enhancing the overall effectiveness of sponsorship activities.

## 2. Literature Review and Hypotheses

In sports sponsorship research, understanding consumers’ reactions to sponsorship activities is the key to enhancing sponsorship effectiveness. Consumers’ sports involvement and the fit between the sponsoring brand and the sponsored event are generally considered the two most important factors influencing sponsorship effectiveness. Researchers have conducted extensive studies on these factors and achieved significant findings.

### 2.1. The Impact of Sports Involvement on Consumer Perception

Currently, research on sports involvement primarily focuses on quantitative methods. Researchers have pointed out that sports involvement not only includes consumers’ knowledge and participation levels in specific sports, but also their emotional investment in and identification with sports teams [18]. Existing research findings indicate that consumers’ sports involvement affects the effectiveness of sports sponsorship activities on both cognitive and attitudinal levels [19]. Firstly, on the cognitive level, sports involvement influences consumers’ perceptions of sponsorship activities. This mainly occurs through promoting a deeper understanding of sponsorship information, enhancing the recognition of fit, forming brand associations, evaluating brand credibility, and considering the social environment. Studies have shown that sports involvement enhances consumers’ knowledge in the sports field, leading those with sports involvement to have a more accurate understanding of sponsorship information, including the identity of the sponsors and the nature of the sponsorship activities [20]. Enhanced cognitive abilities enable consumers to more easily recognize and evaluate the fit between sponsors and sports events. Given that the level of fit directly impacts consumers’ understanding of sponsorship activities, a high fit between sponsors and sports events can foster positive brand images and increase brand awareness [21]. Moreover, consumers with sports involvement tend to form brand associations through the connection between sponsorship activities and their favorite sports events, and these associations may influence future purchase decisions [22]. Additionally, the rich knowledge base brought about by sports involvement leads them to show greater detail in evaluating the credibility of sponsors, thoroughly considering the authenticity of sponsorship activities and the intentions of sponsors [23]. Ultimately, consumers’ sports involvement promotes a deeper understanding of social influence and group norms, enabling them to accurately assess the popularity of sponsorship activities within their social circles and sports communities [24]. Therefore, sports involvement significantly impacts consumers’ cognition of sponsorship activities.

### 2.2. The Impact of Sports Involvement on Consumer Attitudes

Similarly, on the attitudinal level, consumers’ sports involvement multidimensionally affects their attitudes towards sponsorship activities, including aspects such as identification, fit, emotional investment, and group emotions [25]. Firstly, research indicates that sports involvement enhances consumers’ identification with brands related to their favorite sports or teams, viewing sponsorship as support for their personal preferences, which fosters positive attitudes [26]. Secondly, when consumers perceive a high degree of fit between sponsors and events, meaning the sponsorship brand is closely associated with their beloved sports activities, their brand attitude and purchase intention are enhanced due to the perceived support for their interests and values by the sponsor [27,28]. Additionally, emotional investment is a key factor. Consumers with high sports involvement typically have strong emotional investments in the activities they participate in, and this emotional investment can translate into support for the sponsoring brand, especially when the sponsorship activities elicit positive emotions related to sports participation [29]. Finally, group influence plays a crucial role in the impact of sports involvement on consumers. Highly involved sports consumers usually have a relatively stable sports social network (e.g., a football team’s fan club) and, thus, their attitudes towards sponsors are inevitably influenced by this group, whereas consumers with lower sports involvement are less affected by such influences [30,31]. Combining these factors, consumers’ sports involvement shapes their attitudes towards sponsorship activities through their perception of identification, fit, emotional investment, and social influence [32].

Finally, it is essential to distinguish between sports involvement and brand familiarity. Although both brand familiarity and involvement significantly influence consumer behavior, they differ in their concepts and mechanisms of action. Brand familiarity is the extent of recognition and knowledge consumers have about a brand, typically derived from past interactions and experiences. Consumers with high brand familiarity tend to process brand-related information quickly and automatically, relying on memory and experience rather than ‘#analysis. In contrast, involvement refers to the level of interest and psychological engagement consumers have with a brand or product in a given context. Consumers with high involvement are more likely to engage in comprehensive information processing, analyze brand information carefully, and develop attitudes and behaviors accordingly.

This study examines involvement, investigating how consumers process brand information in high-involvement situations and how this cognitive processing affects brand attitudes and purchase intentions. In sports consumption, involvement is more significant than brand familiarity. Sports consumption frequently involves strong personal interest, emotional investment, and identification with particular sports or brands. This high level of involvement directly impacts consumers’ attitudes toward the brand, their loyalty, and their purchasing decisions. Unlike other consumption domains, sports consumption is not solely driven by product functionality or brand familiarity, but is deeply rooted in personal interest and emotional connection. Consumers with high involvement in sports often evaluate how well the brand aligns with their values, interests, and active lifestyle, rather than focusing solely on the brand’s visibility or familiarity. Research indicates that in high-involvement situations, consumers are more likely to engage in thorough information processing and base their purchasing decisions on this cognitive process [33]. In sports consumption, this thorough cognitive processing is particularly important as it directly influences the formation of brand attitudes and behavioral choices. Therefore, this study uses involvement as the primary variable to capture the cognitive processing and decision-making mechanisms of consumers in sports consumption more accurately. This approach enables our research to reveal consumer behavior motivations in sports consumption contexts more effectively and offers targeted insights for brands developing sports marketing strategies.

### 2.3. The Role of Brand Fit in Sports Sponsorship

Building on this theory, Lacey and Close (2013) [34] proposed a framework based on consistency to guide service brand sponsors in their event sponsorship decisions. Their findings elucidated the importance of alignment between the event and the sponsor and how this alignment enhances key outcomes in consumer relationships. Thus, the match between the sponsor brand and sports events is now considered a crucial factor influencing the effectiveness of sponsorship. Research by Kwon, Ratneshwar, and Kim (2016) [35] confirms that brand sponsorships can enhance the consistency of brand images with sports events, especially when there is high functional similarity. This suggests that consumers are more likely to perceive the brand as a natural extension of the sports event when the brand’s products or services are highly relevant to the event, thereby enhancing positive brand image and identification. Additionally, Aguiló-Lemoine, Rejón-Guardia, and García-Sastre (2020) [36] found that on sports event websites, the fit between sponsorship brands and events significantly enhances sponsorship effectiveness. Specifically, exposure of high-fit brands on event websites effectively captures consumers’ attention and increases their interest in and liking of the brand. Research by Brochado, Dionísio, and Leal (2018) [37] emphasizes the importance of alignment between brands and national football teams in enhancing brand image. They find that when brands align closely with the values and images of national teams, consumers are more likely to support and endorse the brand. This consistency in values and image increases brand affinity and credibility. Furthermore, Park and Sihombing (2020) [38] demonstrate that high fit between sponsors and events not only enhances brand image, but also significantly improves consumer attitudes towards the brand. Specifically, when consumers perceive a high degree of match between the brand and the event, they are more likely to develop positive emotions towards the brand and prioritize it in future purchasing decisions. Devlin and Billings (2018) [39] further illustrate the significant impact of sponsorship consistency with events on brand awareness and attitudes. They found that when brands align culturally and in terms of values with events, consumers have more positive perceptions of the brand, leading to enhanced market performance. Research by Cui, Lee, and Jin (2019) provides another perspective, showing that high consistency between sponsors and events helps mitigate negative news impacts on brands, enhancing brand attitudes and purchase intentions. Specifically, high-fit sponsorship activities can provide a protective effect for brands during negative news incidents, maintaining consumers’ trust in and favorable attitudes towards the brand. These studies collectively highlight the crucial role of high fit between sponsorship brands and sports events in enhancing brand image, improving consumer attitudes, and increasing purchase intentions. Therefore, when selecting sports sponsorship opportunities, brands should pay particular attention to alignment with sports events to maximize sponsorship effectiveness.

In conclusion, sports involvement and brand fit are critical factors influencing the effectiveness of sports sponsorships. To further explore their interrelationship, the Theory of Perceived Fit provides a significant framework. Originating from consumer behavior, this theory was first proposed by Bettman (1979) [40] to explain the psychological processes consumers undergo when selecting and evaluating products, centered around the core concept of “fit”, which assesses how well consumers subjectively perceive that an option aligns with their preferences, needs, and expectations. As the theory has evolved, it has been widely applied in fields such as marketing, brand management, and advertising strategies, serving as a crucial tool for understanding consumer responses and behaviors. Its applications span product selection, brand evaluation, advertising effectiveness, and brand extension strategies. In sports marketing, the Theory of Perceived Fit is extensively utilized to explain and optimize sports sponsorship strategies. For sports sponsorships, perceived fit primarily hinges on the alignment between the brand and the sponsored sports event, as well as the level of consumer involvement in the sport [41]. Sports involvement reflects consumers’ interest and participation in sports activities, while brand fit denotes the degree of alignment between the brand and the sports event. Research indicates that higher brand fit with sports events strengthens consumers’ brand identification and favorability. Concurrently, consumers’ sports involvement, indicating their interest and engagement in specific sports, also significantly influences sponsorship effects, with highly involved consumers exhibiting greater acceptance and loyalty towards sponsor brands. This study aims to integrate the Theory of Perceived Fit and delve into the specific impacts of these two concepts on consumer cognition and attitudes across different contexts, explaining experimental results and providing a scientific basis for optimizing sports sponsorship strategies. According to the Theory of Perceived Fit, existing research has extensively explored how sports involvement influences consumer perceptions and attitudes towards sponsorship activities, highlighting the advantages of highly involved consumers in understanding and evaluating sponsorship information. However, there remains a significant research gap concerning the interactive effects of sports involvement and brand fit on sponsorship effectiveness. While existing studies primarily employ quantitative analysis to explore the impact of sports involvement on sponsorship effects, there is limited research on how consumers with varying levels of sports involvement process sponsorship information at psychological and neurobiological levels, particularly regarding emotional responses, memory encoding, and brand attitude formation. Furthermore, although studies confirm that differences in sports involvement influence consumer cognition and attitudes towards sponsorships with varying degrees of fit, the underlying neural mechanisms and principles remain unclear. These research gaps may hinder sponsors’ ability to accurately identify which activities effectively enhance brand awareness and image among their target customer base, making it challenging to assess the effectiveness of sponsorship activities accurately and potentially leading to significant wastage of marketing resources. Therefore, by integrating the Theory of Perceived Fit, this study will explore the interaction between sports involvement and brand fit and their profound impacts on consumer cognition and attitudes, thereby providing theoretical support to enhance the scientific rigor and effectiveness of sports sponsorship strategies.

This study aims to explore, through the introduction of Predictive Coding Theory, how consumers with different levels of sports involvement process sponsorship information at the psychological and neurobiological levels. Predictive Coding Theory explains how the brain processes information, emphasizing that the brain constructs internal models based on long-term accumulated experience and knowledge to predict external events, and continually adjusts and improves these predictions based on the discrepancies (prediction errors) between predicted and actual sensory inputs [42]. Initially proposed by Friston (2005) [15], Predictive Coding Theory has been widely applied to explain perceptual and cognitive processes [43]. The theory posits that the brain actively predicts external events rather than passively receiving information, comparing these predictions with actual sensory inputs [44]. Through this ongoing process of prediction and refinement, the brain can effectively handle complex information inputs. Additionally, this theory suggests that the nervous system can make anticipatory predictions about incoming information, revealing the neural computations underlying theory of mind [45]. The theory underscores that the brain’s predictive capability relies on past experiences and knowledge, which are used to construct internal models that facilitate rapid responses when encountering new information [46]. In marketing and brand sponsorship research, Predictive Coding Theory is employed to understand how consumers process brand information based on past experiences and knowledge. For instance, when consumers see a brand sponsoring a sports event, their brains predict the association between the brand and the event based on previous experiences. If this association aligns with their predictions, their cognition and attitudes towards the brand are enhanced; otherwise, they may experience cognitive conflict, requiring more cognitive resources to resolve such inconsistencies [47].

Predictive Coding Theory posits that the brain continuously generates predictions about sensory inputs and adjusts these predictions based on actual sensory information to minimize prediction error. This theory suggests that individuals with high sports involvement, who possess extensive knowledge on and a strong interest in sports, develop more complex predictive models for processing brand information. Consequently, even when encountering a sponsor brand mismatch, these individuals may reduce prediction errors through internal cognitive adjustment mechanisms, resulting in less pronounced differences in N270 and LPP amplitudes between match and mismatch conditions. This implies that their cognitive conflict is less pronounced because their internal models are adept at reconciling discrepancies. In contrast, individuals with low sports involvement, who have less knowledge and simpler predictive models regarding the sports domain, experience larger prediction errors when faced with sponsor brand mismatches. Their internal mechanisms are less effective at correcting these errors, leading to significantly increased N270 and LPP amplitudes under mismatch conditions, indicating greater cognitive conflict and difficulty in processing the sponsor information. This study applies Predictive Coding Theory to explore how consumers predict and process sponsorship information based on their level of sports involvement and further analyzes how these predictions influence their perceptions and attitudes towards sponsorship–event fit. According to the theory, consumers’ sports involvement—comprising their sports knowledge and participation experiences—shapes their internal models for interpreting sports sponsorship activities. Highly sports-involved consumers, with their detailed internal models, experience less cognitive and emotional conflict when encountering relevant sponsorships, leading to more positive responses. In contrast, those with low sports involvement, due to simpler internal models, face greater cognitive and emotional challenges when processing sports sponsorship information, which may result in more negative responses. By understanding how consumers with varying levels of sports involvement process information at a neurobiological level, this study not only provides a scientific basis for sports brands in selecting and executing sponsorship activities, but also offers new insights for evaluating and optimizing sponsorship effectiveness. Brands can leverage these findings to develop more precise marketing strategies and enhance the overall impact of their sponsorship efforts.

In this context, ERP (event-related potential) technology provides an effective method to directly measure brain activity, in order to delve into how consumers, based on their varying levels of involvement, process sponsorship information at the psychological and neuroscientific levels. In cognitive neuroscience, numerous studies have demonstrated that specific brain electrical signals reflect individuals’ cognitive and emotional states in relation to sponsored advertisements when watching sports events [48,49]. Additionally, the use of event-related potential (ERP) offers a convenient, non-invasive, objective, and high-temporal-resolution method that avoids the subjectivity inherent in self-report methods such as questionnaires. Alonso Dos Santos et al. (2023) [49] used EEG and self-reports to study sponsorship consistency, aiming to measure sponsorship effects at a physiological level. EEG, which was first demonstrated in 1875 by British physiologist Richard Caton [50], detects the brain’s spontaneous electrical oscillations. In this study, ERP (event-related potential) combined computer averaging techniques to enhance electrical signals induced by stimulus events, creating a new signal that directly reflects cortical responses to emotion, perception, and cognition with high temporal resolution. ERP, as it is now known, can be defined as the electrical changes in brain regions occurring in response to or withdrawal of stimuli, or when specific psychological factors are present [51]. ERP has also been widely applied in consumer-decision-making research. For example, Zhang (2022, 2024) [52,53] investigated how framing effects influence green product purchasing decisions, Ozkara (2021) [54] examined ERP changes in consumer decision-making under conscious and unconscious conditions, and Shang (2024) [55] used ERP to measure changes in consumer acceptance in response to AI recommendations and different product types. In this context, ERP technology provides a valuable tool for exploring how consumers process sponsorship information at psychological and neuroscientific levels across different levels of involvement. Research indicates that specific brain electrical signals can reflect individuals’ cognitive and emotional responses to sponsorship advertisements during events. By employing ERP, researchers can avoid the subjectivity of traditional self-report methods and directly measure brain activity to gain deeper insights into consumer decision-making processes and responses to sponsorship advertisements.

This study primarily focuses on two components: N270 and LPP. The N270 component, a negative wave peak occurring at approximately 270 milliseconds, is triggered by conflicts between external stimuli and internally generated information during cognitive tasks (Shi et al., 2005) [56]. These conflicts range from simple attributes such as color, size, shape, and spatial position to complex stimuli such as face recognition using face photographs. The amplitude of N270 is related to the degree of information conflict encountered by individuals in cognitive tasks. Specifically, when information that individuals need to process conflicts with their internal models or expectations, the amplitude of N270 increases. The Late Positive Potential (LPP), appearing in a later time window, is a stable electroencephalographic component that characterizes the formation of subjects’ attitudes and is used to validate attitude evaluations after stimulus presentation [57,58]. The amplitude of LPP typically increases when individuals are presented with stimuli of high emotional value or particular relevance, reflecting sustained attention towards and deeper processing of these stimuli. This suggests that when stimuli touch upon individuals’ emotions or hold specific meaning for them, the brain allocates more resources to process this information, resulting in larger LPP amplitudes [59,60]. Therefore, by designing EEG experiments, we can observe differences in N270 and LPP amplitudes among consumers with varying levels of sports involvement when confronted with sports events and sponsorship brands of different fit levels. This approach allows us to investigate how different levels of sports involvement and sponsorship fit influence consumers’ cognition and attitudes towards sponsorship activities.

In summary, this study proposes the following hypotheses:

**H1.** 
*Based on Predictive Coding Theory, consumers with varying levels of sports involvement develop different internal models for processing sports-related information, leading to cognitive differences when encountering sponsorship information with varying brand fit levels. Specifically, it is hypothesized that participants with high sports involvement will have higher N270 amplitudes for sponsorship brand mismatches compared to brand matches. Similarly, it is hypothesized that this phenomenon will also be observed in the low sports involvement group.*


**H2.** 
*Consumers with varying levels of sports involvement are hypothesized to show differences in attitudes toward sponsorship information with different brand fit levels. Specifically, it is hypothesized that participants with high sports involvement will exhibit differences in LPP amplitude, with higher LPP amplitudes observed for sponsorship brand mismatches compared to brand matches. This phenomenon is also expected to be present among participants with low sports involvement. I agree with the editor’s suggestion and have made revisions in H1 and H2.*


**H3.** 
*According to Predictive Coding Theory, the degree of brand fit in sports sponsorship influences the extent of cognitive and emotional responses differently for consumers with varying levels of sports involvement.*


## 3. Experiment Methodology

The experiment employed a 2 (High Sports Involvement, Low Sports Involvement) × 2 (Matched Condition, Unmatched Condition) design, recruiting a total of 60 healthy volunteers aged between 18 and 35 years. During the experiment, three participants exhibited excessive EEG artifacts, so they were excluded from the final data analysis, leaving a total of 57 valid participants (data analysis conducted accordingly). The average age of the participants was 23 years (M = 23.06, SD = 3.53), consisting of 30 males and 27 females. To ensure the reliability of the experimental data, all participants were right-handed, without any psychological or psychiatric disorders, and had normal or corrected-to-normal vision. According to these criteria, participants were categorized into a high-involvement group (28 participants) and a low-involvement group (29 participants).

### 3.1. Sports Involvement Grouping

Sports involvement (Sport Involvement) refers to an individual’s interest, attention, and participation level in sports activities. It reflects the amount of time, energy, and emotional investment an individual puts into sports. Sports involvement is not only evident in one’s passion for and frequent participation in a particular sport, but also includes emotional attachment to sports, the significance of sports in one’s life, and the symbolic value of sports to one’s identity.

In current research in this field, one of the widely used questionnaires is the Sport Fan Motivation Scale (SFMS) developed by Funk (2001) [61]. This questionnaire includes multiple dimensions and items such as interest, emotional attachment, participation frequency, centrality, and symbolic value, which comprehensively assess consumers’ sports involvement.

In this study, to accurately differentiate participants based on their sports involvement, researchers distributed the SFMS questionnaire before the experiment. Participants rated various items related to sports involvement on a 5-point scale, where 1 indicates strongly disagree and 5 indicates strongly agree. Subsequently, statistical analysis was performed on the questionnaire data to calculate each participant’s total score for each dimension. Based on the total scores across these dimensions, participants were categorized into high sports involvement and low sports involvement groups. According to the criteria mentioned above, the study ultimately categorized participants into a high-involvement group (28 individuals) and a low-involvement group (29 individuals).

### 3.2. Sponsor Brand Grouping

From the ratings and sales figures on major domestic online shopping platforms (such as Tmall and JD.com), the top twenty well-known sports brands were selected. Additionally, based on the level of engagement and popularity on Weibo, the top ten sports disciplines were identified. A familiarity survey was then conducted to assess familiarity with these sports brands and disciplines, using a 5-point scale, where “1” indicated unfamiliarity, “2” somewhat unfamiliar, “3” neutral, “4” familiar, and “5” very familiar. Ultimately, 55 valid surveys were collected, comprising 33 males and 22 females, with ages ranging from 18 to 35 years: in total, 39 participants were aged 18–25, 13 were aged 25–30, and 3 were over 30. The survey revealed the top ten well-known sports brands (Nike, Adidas, Under Armour, Puma, Li-Ning, Anta, Decathlon, Mizuno, Yonex, Reebok) and their corresponding popular sports disciplines (basketball, soccer, badminton, volleyball, table tennis, mountaineering, running, boxing). In addition, a control group of ten unknown sports brands (Theo, Nick, Belock, Shufei, Haosha, Lash, Bart, Senqiong, Diado, Clauf) was included. Although these brands are imaginary and do not exist in reality, participants were informed before the experiment began that these brands were real but less known, to serve as distractors in their judgments. All materials used in the experiment are summarized in Table 1.

On the other hand, to ensure the reliability of the experimental results, we implemented several measures in this study to effectively control for potential interference from brand familiarity. First, we deliberately selected brands with lower consumer familiarity to reduce the potential automated processing effects associated with highly familiar brands, thereby ensuring more precise measurement of the impact of involvement on cognitive processing. This approach aligns with previous research, demonstrating that selecting less familiar brands effectively reduces brand familiarity interference, allowing for a better analysis of the influence of involvement on consumer cognitive behavior (Ma et al., 2020) [62]. Additionally, in the preliminary stage of the experiment, we assessed participants’ familiarity with each brand through a survey, ensuring that the brands selected for the experiment did not show significant familiarity differences among participants. This measure minimized the influence of brand familiarity, allowing for a more accurate evaluation of the roles of involvement and brand fit in consumer cognitive processing (Türkel et al., 2016) [63]. Finally, during data analysis, we controlled for brand familiarity as a covariate, ensuring that the main effects of involvement and brand fit were clearly exhibited, free from the interference of familiarity. The literature indicates that brand familiarity and involvement affect consumer cognition through different mechanisms, making it necessary and effective to control for brand familiarity (Lim and Chung, 2014) [64]. Through these experimental design and data analysis controls, we are confident that the interference from brand familiarity has been effectively eliminated, fully validating the impact of involvement.

Matching and non-matching conditions between sports brands and sports activities were determined through a survey distributed to 60 college students. Each question in the survey, designed as multiple-choice, pertained to the association of each brand with 8 specific sports activities. Following Keller (1993) [65], brand association is defined as consumers’ spontaneous retrieval of brand information stored in memory, where selected nodes largely influence consumer purchase intentions. The questionnaire used a Likert 5-point scale for scoring: 5 for very matching, 4 for moderately matching, 3 for somewhat matching, 2 for weakly matching, and 1 for not matching at all. Test samples with mean scores above 2.8 were considered matching conditions, while those below 2.8 were deemed non-matching conditions, as shown in Table 2 and previous research.

Based on the initial survey results regarding consumer matching between sports brands and sports activities, the experimental materials were categorized into matching conditions, non-matching conditions, and control conditions, as shown in Table 3.

Based on the results of the sports activities and sports brands matching survey above, it is evident that each sports brand has corresponding sports activities. Some matching combinations (e.g., Nike and basketball, Adidas and soccer) align with our case analyses discussed in the Literature Review section. However, other matching combinations (e.g., Li-Ning and mountaineering, Anta and badminton, Mizuno and table tennis) were also found to have some form of matching relationship. This suggests that sports activities are significant factors influencing consumer brand perception, thereby affirming the effectiveness of this study in exploring the alignment between activities and brands.

### 3.3. Experimental Design Procedure

The study on sponsorship effect was conducted using ERP. In later stages of data analysis, a combination of SPSS 26.0 for behavioral data analysis and Curry 8.0 for EEG data analysis was employed. This research adopted a “prime–probe” paradigm with two stimuli to simulate real purchasing scenarios, aiming to explore decision mechanisms in the alignment judgment between sports activities and consumer brand cognition. The experiment utilized a “prime–probe” paradigm (Stimulus 1–Stimulus 2), where the prime stimulus was the sports brand name and the probe stimulus followed with the sports activity name. This paradigm was used to detect consumer evaluations of and attitudes toward sports brands following the presentation of the brand name. The experiment took place in a quiet laboratory environment with a voltage not exceeding 220 V. Participants sat comfortably in soft leather chairs with back support to prevent fatigue during the experiment, and a chin rest was provided to stabilize the head position. Prior to the experiment, participants signed confidentiality agreements, received explanations about the experimental procedure and requirements, and underwent a brief pre-experiment session before the formal start. Based on the aforementioned experimental design procedure, the sequence and duration of experimental stimuli are illustrated in Figure 1.

## 4. Experimental Data Analysis and Organization

### 4.1. Behavioral Data Analysis

In Figure 2, the boxplot of purchase rates shows that the high-involvement group exhibited significantly higher purchase rates under both matching and non-matching conditions compared to the low-involvement group. The inter-group comparisons reveal a notable internal difference between the high-involvement group with a sports background (M = 63.03%, SD = 23.18%) and the low-involvement group without a sports background (M = 43.51%, SD = 25.39%) under matching conditions. Similarly, under non-matching conditions, the high-involvement group (M = 46.62%, SD = 24.50%) showed higher purchase rates than the low-involvement group (M = 42.48%, SD = 26.89%).

Next, through comparisons between conditions, it was found that the high-involvement group (M = 63.03%, SD = 23.18%) and the low-involvement group (M = 43.51%, SD = 25.39%) exhibited higher purchase rates under matching conditions compared to non-matching conditions (M = 46.62%, SD = 24.50%) and (M = 42.48%, SD = 26.89%) (see Table 4). We found no significant main effect between the participant categories, indicating that the high and low-involvement groups did not differ significantly (F(1,57) = 0.541, *p* = 0.466). However, there was a significant main effect between the condition categories, showing higher purchase rates under matching conditions compared to non-matching conditions (F(2,57) = 82.077, *p* < 0.001). There was also a significant interaction effect between participant categories and condition categories, indicating that the high and low-involvement groups significantly differed in their responses to matching and non-matching conditions (F(2,57) = 10.474, *p* = 0.002). A post hoc analysis of the interaction effect revealed that for matching conditions, the high-involvement group was significantly higher than the low-involvement group (*p* = 0.014); however, there were no significant differences between the groups under the other two conditions. For consumers in the low-involvement group, the purchase rates were significant under both matching and non-matching conditions.

As shown in Figure 3 and Table 5, in terms of reaction times, participants with high sports involvement exhibited longer reaction times than those with low sports involvement, regardless of whether the brand matched the sports activity. This result also indicates the existence of different predictive coding systems between the two groups.

The results indicate a non-significant main effect between the participant groups (F(1,57) = 1.806, *p* = 0.187), a significant main effect between the conditions (F(2,57) = 39.438, *p* < 0.001), and a significant interaction effect (F(2,57) = 4.225, *p* = 0.034). Based on the behavioral data results, there were no significant differences between the groups in purchase rates and reaction times. However, the descriptive data show that under both matching and non-matching conditions, the high sports involvement group exhibited higher purchase rates compared to the low sports involvement group, possibly due to the high demand for sports products among the high-involvement group, who are primary consumers of sports products. Furthermore, in the comparisons between the conditions, all the participants showed higher purchase rates under matching conditions compared to non-matching conditions. This suggests that a higher alignment between sports activities and sports brands in matching conditions positively influences consumer purchase intentions and, consequently, increases purchase rates.

### 4.2. EEG Data Analysis

#### 4.2.1. EEG Data Analysis Steps

The process of organizing and analyzing the collected EEG data using the Scan4.3 EEG recording system is as follows:Checking of EEG data: The total experiment duration is approximately 14 min. During the data recording, participants may move slightly, clench their teeth, or swallow saliva, which could lead to muscle artifacts and data deviations. It is necessary to inspect and remove segments with significant fluctuations or deviations from the EEG data.Removal of unnecessary electrodes: This includes eye movement electrodes, CB1, and CB2. To correct for eye movement artifacts, which are inevitable during the experiment, vertical eye movement (VEO) electrodes are used as reference electrodes.Digital filtering: A bandpass filter of 0.1–30 Hz is applied to remove powerline interference (48–52 Hz, 98–102 Hz). Filtering not only eliminates noise from the data recording, but also filters out EEG data frequencies irrelevant to the experiment, resulting in smoother final data.Segment processing: EEG data are recorded continuously and include stimulus marks, intervals between marks with EEG information, and irrelevant signals during inter-block rests. Based on the experimental objectives, EEG data corresponding to stimulus marks under the same conditions are segmented. For this experiment, each complete mark process spans the period from 200 ms before stimulus onset to 1000 ms after presentation, which is essential for subsequent averaging.Interpolation of bad electrodes and discarding of bad segment processing: according to neuroscientific standards and the experimental design, data accuracy and precision can be ensured.ICA artifact removal: To ensure the reliability of the final experimental data, artifacts such as horizontal eye movements (HEOR), vertical eye movements (VEOR), left ear (M1), right ear (M2) and other recorded artifact electrodes are removed. Finally, EEG data with amplitudes beyond ±80 μV are excluded.Data export: Select and define ERP components, export ERP waveforms from the target electrode sites, construct global topographical maps under various conditions, and observe cognitive neural activity related to specific stimuli. Export the averaged amplitude values of relevant ERP components during corresponding time segments for statistical analysis. Use repeated-measures ANOVA to analyze data significance.Data analysis processing: Initially, perform within-subject averaging across all segments, followed by averaging across subjects within the same group. Extract the average wave amplitudes of both groups (high sports involvement group and low sports involvement group) and both conditions (matching and non-matching) for e-measures ANOVA. Finally, conduct post hoc analyses for significant results (*p* < 0.05). Correct the degrees of freedom and *p*-values using the Greenhouse–Geisser correction for statistics that do not meet sphericity assumptions, and apply Bonferroni correction for post hoc comparisons.

#### 4.2.2. EEG Component Analysis

The EEG data analysis employed in this study utilized a 1000 ms time window. Following the “cue–probe” experimental paradigm design, participants from different backgrounds (high sports involvement group/low sports involvement group) were subjected to different conditions (matching/non-matching), resulting in six distinct EEG waveforms. Given that ERP experiments often require the selection of specific time windows to analyze significant component differences, this study integrated findings from extensive scholarly research and experimental designs to finalize these six different EEG waveforms. Ultimately, this study focused on four EEG components: P2, N270, P300, and LPP. The statistical analysis of the EEG data was conducted using repeated-measures ANOVA with SPSS 26.0. Next, detailed analyses of each component will be presented.

##### N270 Component

According to the previous literature on the N270 component and the hypotheses proposed in this study, the time window selected for this component was 260–340 ms. Consistent with the aforementioned component analysis approach, a statistical analysis was conducted on six electrode points: F3, FZ, F4, FC3, FCZ, and FC4. The average amplitude of the N270 component was subjected to a (2 × 2) repeated-measures ANOVA based on these six electrode points.

The analysis revealed significant main effects between the two conditions (matching, non-matching), with F(1,57) = 3.757, *p* = 0.028. However, there was no significant between-group effect between the high sports involvement group and the low sports involvement group, with F(2,57) = 0.093, *p* = 0.763, nor was there a significant interaction effect between group and condition, with F(1,57) = 0.801, *p* = 0.428. The effect sizes, measured as partial η², were as follows: the main effect of condition (matching vs. non-matching) is 0.062, and the main effect of group (high vs. low sports involvement) is 0.003, and the interaction effect between the group and the condition is 0.014. These values indicate that the effect size for the condition’s main effect (matching vs. non-matching) was substantial, while the effect sizes for the other two effects were very small.

The post hoc analysis indicated that both groups of participants showed significant differences between matching and non-matching conditions (*p* < 0.001). Integrating these findings, as depicted in Figure 4, although the similarity matching differences in information processing between the high sports involvement group and the low sports involvement group were not significant, all the participants exhibited significant differences in similarity matching between matching and non-matching conditions. Specifically, all the participants showed significantly more pronounced N270 peaks under non-matching conditions compared to matching conditions, indicating that under non-matching conditions, the participants’ brainwave amplitudes were more intense.

Based on the topographical maps of the N270 components, we can visually observe the whole-brain distribution and activation levels of the high sports involvement group and low sports involvement group under matching and mismatching conditions, as shown in Figure 5.

##### LPP Component

According to the previous literature on LPP components and the hypotheses proposed in this study, we selected the late time window of 400–500 ms. Following the same component analysis method as mentioned above, we conducted a statistical analysis using six electrodes: F5, F3, F1, FC5, FC3, and FC1 (see Figure 6). The average amplitude of the LPP component was analyzed using a repeated-measures ANOVA based on a (2 × 2) design across these six electrodes.

The results of the analysis indicated a significant main effect between the two conditions (matching, mismatching), with F(1,57) = 10.060, *p* < 0.001. However, there was no significant between-group effect between the high sports involvement group and the low sports involvement group, with F(2,57) = 1.100, *p* = 0.301, nor was there a significant interaction effect between group and condition, with F(1,57) = 0.456, *p* = 0.636. The effect sizes, measured as partial η², were as follows: the main effect of condition (matching vs. non-matching) was 0.150, the main effect of group (high vs. low sports involvement) was 0.037, and the interaction effect between group and condition was 0.008. These values indicate that brand fit has a significant effect on the LPP component, suggesting that the degree of brand match has a moderate impact on cognitive processing. The effect of sports involvement on the LPP component was relatively small, indicating no significant differences in neural responses to brand matching between the high- and low-involvement participants. The interaction effect between group and condition on the LPP component was negligible, suggesting that there is no significant interaction between the high and low-involvement consumers’ responses to brand match and mismatch conditions.

The post hoc analysis revealed that both groups of participants performed significantly better under matching conditions compared to mismatching conditions (*p* < 0.001). Integrating these findings, as depicted in Figure 7, while there were no significant differences between the high sports involvement group and the low sports involvement group in decision-making during information processing, significant differences existed in their decision-making under matching and mismatching conditions. Specifically, all the participants showed larger LPP amplitudes under mismatching conditions between sports activities and sports brands compared to matching conditions.

Based on the topographical maps of the LPP component, we can visually observe the whole-brain distribution and activation levels of the high sports involvement group and the low sports involvement group under matching and mismatching conditions, as shown in Figure 7.

## 5. Discussion

### 5.1. Discussion of Behavioral Data Results

During the experiment, the participants were required to make rapid decisions. When the participants observed associations between sporting events and matching or mismatching sports brands, the behavioral acceptance rates showed distinct results. The participants exhibited significantly higher acceptance rates for sports brands highly matched with sporting events compared to those with lower matching, indicating the significant impact of consistency between sporting events and sponsored brands on sponsorship effect. Regardless of the audience type, sponsorship outcomes involving mismatched sports brands performed poorly. This underscores the importance of consistency between sporting events and sponsored brands. Specifically, under conditions in which sports brands were highly matched with sporting events, the participants in the high-involvement group showed higher acceptance rates compared to the low-involvement group. Under mismatched conditions, both groups exhibited similarly low acceptance rates.

### 5.2. Discussion of N270 Component

In the high-involvement participants, there was no significant difference in N270 amplitude between the matching and mismatching conditions; however, for the low-involvement participants, the N270 amplitude was significantly higher under the mismatching conditions compared to the matching conditions. This suggests that low-involvement consumers experience greater cognitive conflict when confronted with brand sponsorship information that does not align with their expectations [66]. This finding validates hypothesis H3 proposed in this study: according to Predictive Coding Theory, the degree of brand fit in sports sponsorship influences the extent of cognitive and emotional responses differently for consumers with varying levels of sports involvement. Additionally, across all the participants, sponsor brands under mismatch conditions resulted in significantly higher N270 amplitudes compared to match conditions, thus supporting hypotheses H1 of this experiment.

Predictive Coding Theory posits that the brain continually generates predictions about sensory inputs and adjusts these predictions based on actual sensory inputs, aiming to minimize prediction errors [15]. In the context of sports brand sponsorship, consumers’ brains generate predictions about the relationship between brands and sporting events. When actual sponsorship information matches consumer expectations, prediction errors are minimized. Conversely, mismatches lead to larger prediction errors, resulting in cognitive conflict. Specifically in this study, the high-involvement consumers typically possessed deeper knowledge and stronger interest in the sports domain. According to Predictive Coding Theory, they are likely to have developed more complex and refined prediction models to process brand information related to sports. Therefore, even under conditions of low brand fit, they may reduce prediction errors through internal cognitive adjustment mechanisms, which would explain why there was no significant difference between the matching and mismatching conditions in the high-involvement group.

In contrast, low-involvement consumers may have less knowledge of the sports domain and simpler prediction models. When faced with mismatched brand sponsorship information, their prediction errors are larger, making it difficult to adequately reduce these errors through internal adjustment mechanisms. This explains why, in the low-involvement group, the matching conditions elicited significantly smaller N270 responses compared to mismatching conditions [62].

### 5.3. Discussion of LPP Component

In the field of cognitive neuroscience, the LPP is regarded as brain electrical activity associated with emotional processing, where its amplitude typically reflects the intensity of emotional responses to stimuli and sustained attention. The LPP amplitude in the high-involvement consumer group was significantly lower than in the low-involvement group, a difference observed across the sponsorship activities with varying degrees of brand fit. This suggests that high-involvement consumers may exhibit weaker emotional responses to sponsorship information or lower sustained attention levels towards such information. Additionally, there were no significant overall changes in LPP amplitude between the matching and mismatching conditions, indicating that the general impact on the participants’ emotional response intensity and sustained attention did not differ significantly, contrary to our initial hypothesis.

This result partially validates the hypothesis of this study that participants with different levels of sports involvement would show differences in LPP amplitude when exposed to sponsorship information. However, the lack of significant changes in overall LPP amplitude between matching and mismatching conditions contradicts previous assumptions. According to Predictive Coding Theory, high-involvement consumers demonstrate deeper understanding and interest in specific domains, such as sports activities and sponsorship information. This depth of engagement likely facilitates the formation of more complex and refined prediction models, enabling them to process relevant information more efficiently. Consequently, this efficient information processing may lead to smaller LPP amplitudes in response to known or expected stimuli (whether under matching or mismatching sponsorship conditions), as the brain reduces the generation of prediction errors and the associated allocation of attentional resources.

On the other hand, compared to the low-involvement consumers, the high-involvement consumers generally exhibited lower LPP amplitudes across sponsorship activities, irrespective of whether the sponsorship activities were matched with their respective brands. This suggests that the brains of high-involvement consumers have already optimized the processing of relevant information at a broader level, thereby reducing discriminative responses to specific stimuli under matched conditions. Building upon Kim, Stout, and Cheong’s (2012) [67] findings, we can further understand the efficient processing patterns of high-involvement consumers when dealing with sponsorship information. Their study indicated that consumers’ processing of sponsorship information depends more on available processing resources rather than specific sponsorship content. Therefore, high-involvement consumers tend to employ more efficient processing patterns based on their extensive knowledge and interest in the sports domain rather than the specifics of the sponsorship activities. In summary, the results of this experiment support hypotheses H2.

## 6. Conclusions

This study, based on Predictive Coding Theory, explores the impact of consumer sports involvement on sports brand sponsorship effectiveness, thereby extending the theoretical foundation of the current literature. Previous research has confirmed the importance of sports involvement and brand fit in consumer cognition and emotional responses (e.g., [10]), but few studies have investigated the neural mechanisms underlying their interaction. Previous studies predominantly employed subjective assessment tools, such as self-reports and peer evaluations, to investigate sponsorship effects. These methods are often influenced by participants’ subjective perceptions and evaluation biases. However, this study employed ERP (event-related potential) technology, a neuroscientific method, to investigate sponsorship effects in a more objective and scientific manner. By utilizing ERP technology, this study directly observed the neural responses of consumers when processing brand sponsorship information, thereby reducing the limitations of subjective assessments and providing a more precise evaluation of sponsorship effects. This finding offers a new perspective on sports brand sponsorship strategies, particularly in optimizing brand–consumer fit, and provides a neuroscientific explanation for this interaction. The results indicate that high-involvement consumers possess more complex and refined predictive models in their brains, allowing them to process and interpret brand sponsorship information more efficiently. Thus, high-involvement consumers exhibit higher cognitive and emotional responses regardless of the match between the brand and the sponsorship activity, suggesting that they have optimized their ability to process relevant information at a broader level.

In contrast, low-involvement consumers show significant cognitive conflict (increased N270 amplitude) when faced with mismatched brands and sports activities. This finding challenges the traditional view that low-involvement consumers have lower levels of cognitive processing (e.g., [9]), indicating that even low-involvement consumers generate strong cognitive responses to mismatched information. This provides new insights into how to effectively enhance the response of low-involvement consumers when selecting sponsorship activities. This study not only validates the critical role of brand fit in sponsorship effectiveness according to the perception fit theory, but also reveals the unique mechanisms by which high-involvement consumers process sponsorship information through a neurobiological explanation. The findings also suggest that for high-involvement consumers, sponsorship activities with higher brand fit are more effective, while for low-involvement consumers, enhancing the association between the brand and sports activities can improve sponsorship effectiveness.

However, this study has some limitations. The sample primarily consists of young participants, which may not fully reflect the responses of the general consumer population. Additionally, the experimental design used only ERP technology for measurement, without integrating other physiological indicators and behavioral data, which may have affected the comprehensiveness of the results. Future research could incorporate various neuroscience technologies, integrate behavioral data and other physiological indicators, and further investigate factors such as brand involvement and emotional involvement in the consumer processing of sponsorship information and sponsorship effectiveness. Moreover, expanding the sample size to include participants of different ages and backgrounds would enhance the generalizability and reliability of the results. These findings not only provide theoretical extensions to the existing literature, but also offer scientific evidence for future sports sponsorship strategies, helping sports brands better utilize sponsorship resources and optimize market outcomes. By identifying different response mechanisms for high- and low-involvement consumers, brands can develop more effective differentiated sponsorship strategies, maximizing resource utilization. For high-involvement consumers, brands should focus on selecting highly fitting sports activities to amplify the positive effects of the brand, while for low-involvement consumers, increasing the association between the brand and sports activities can improve their response to the sponsorship activity. This precise strategy not only helps optimize the effectiveness of individual sponsorship activities, but also has a broader impact on the brand’s overall market strategy, particularly in resource allocation and target audience segmentation. Future research could also explore the impact of sponsorship strategies on brand loyalty and brand perception over the long term, providing a more comprehensive understanding of the broader effects of sports sponsorship marketing. The interaction effect between group and condition on the LPP component is minimal, indicating that there is no significant interaction between high- and low-involvement consumers’ responses under matching and mismatching conditions.

## Figures and Tables

**Figure 1 brainsci-14-00940-f001:**
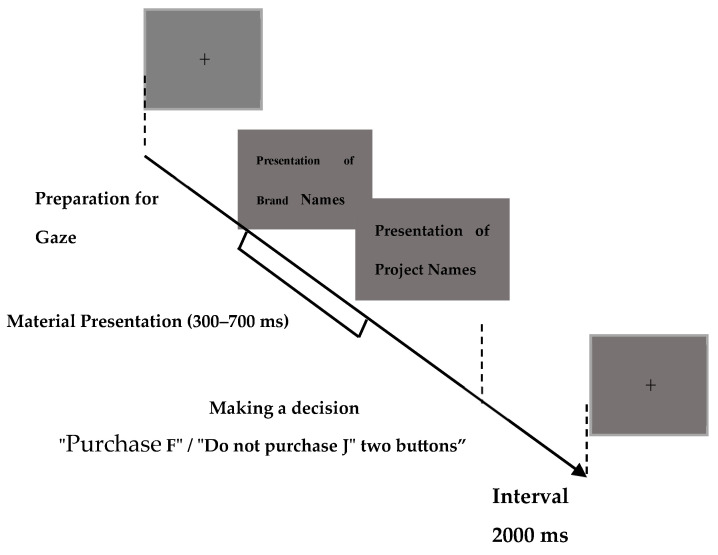
Experimental design procedure.

**Figure 2 brainsci-14-00940-f002:**
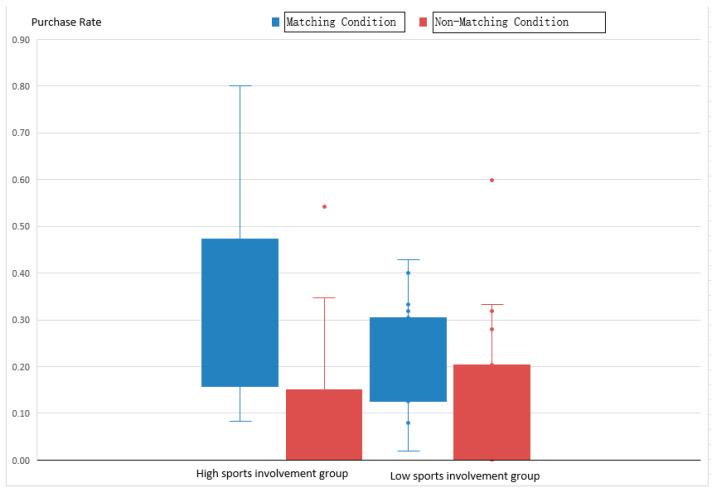
Boxplot of purchase rates.

**Figure 3 brainsci-14-00940-f003:**
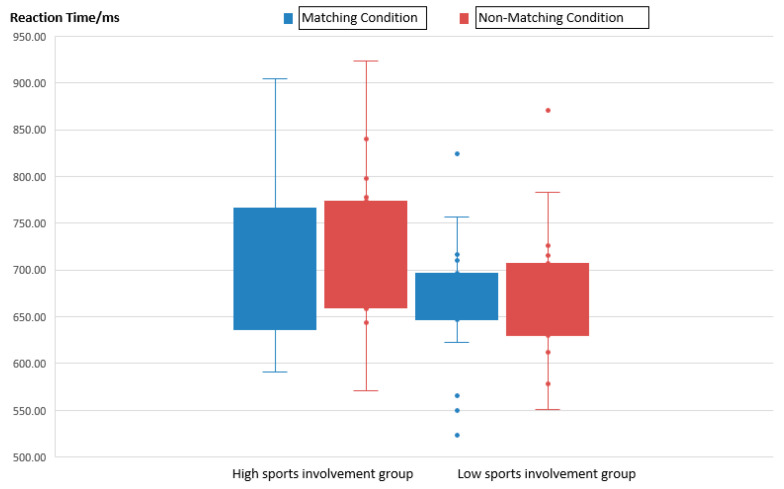
Boxplot of reaction times.

**Figure 4 brainsci-14-00940-f004:**
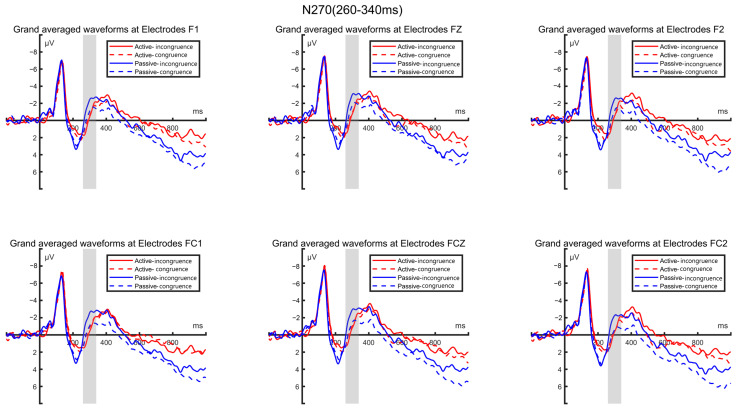
N270 waveforms induced by matching and mismatching conditions in high and low-involvement groups.

**Figure 5 brainsci-14-00940-f005:**
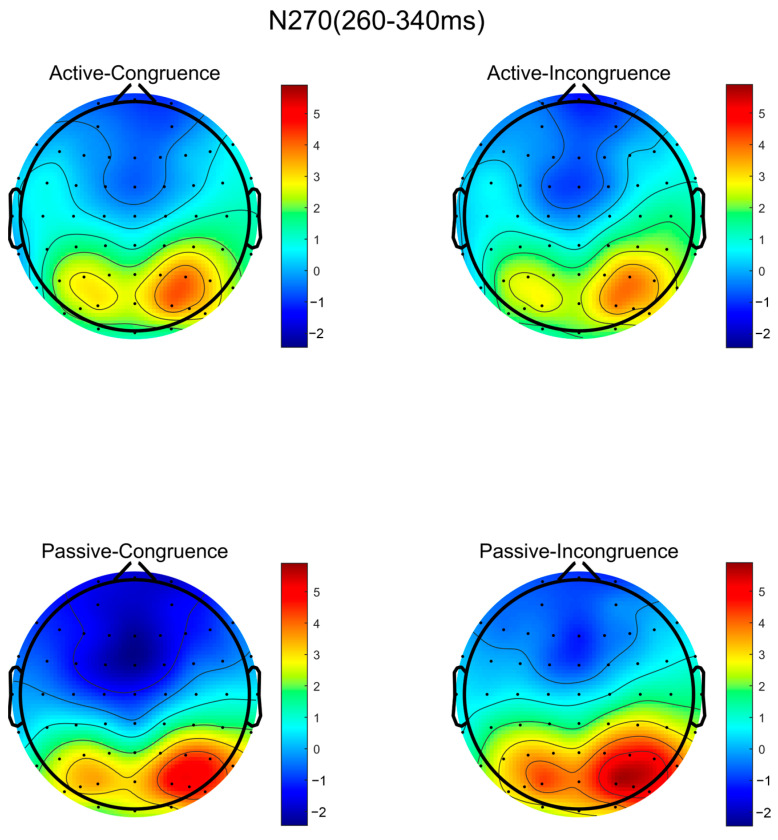
The topographical maps of N270 induced by matching and mismatching conditions in high and low involvement levels.

**Figure 6 brainsci-14-00940-f006:**
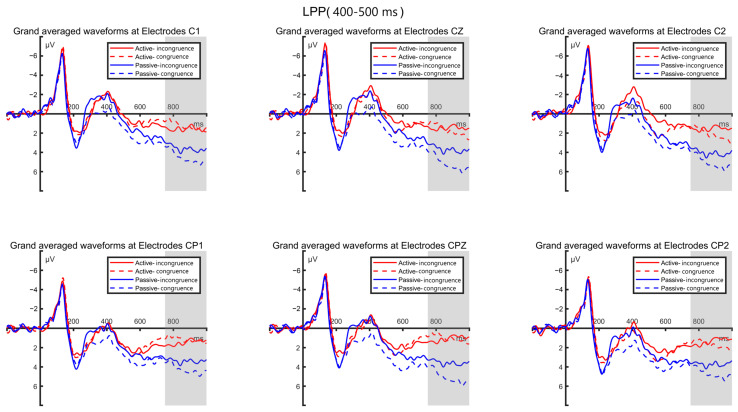
LPP waveforms induced by matching and mismatching conditions in high and low-involvement groups.

**Figure 7 brainsci-14-00940-f007:**
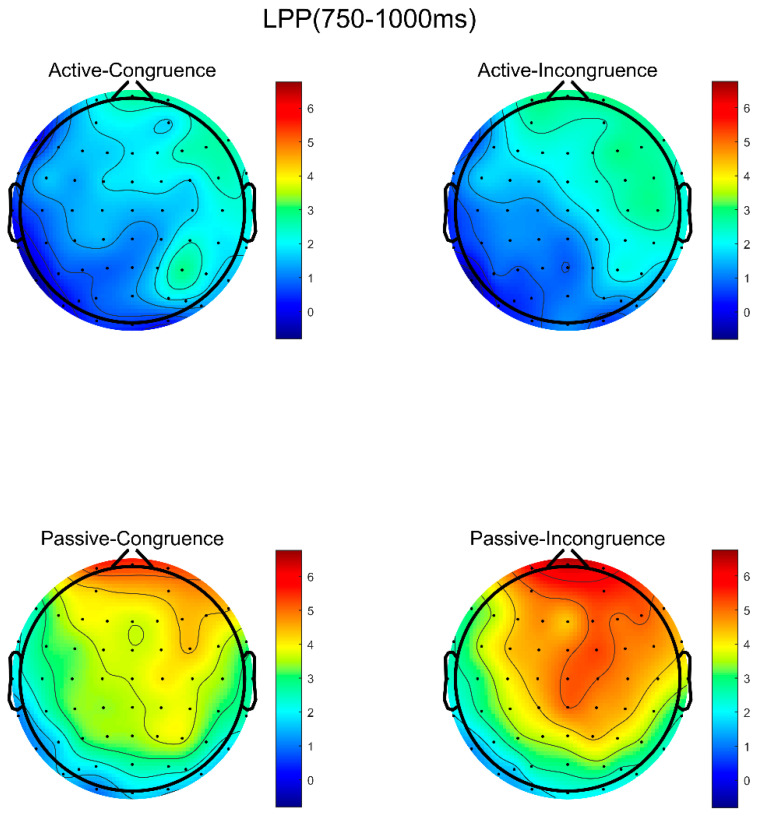
Topographical maps of LPP induced by matching and mismatching conditions in high and low-involvement groups.

**Table 1 brainsci-14-00940-t001:** Sports Brands and Project Names in the Experiment.

**Sports Brands**	Well-known Brands: Nike, Adidas, Under Armour, Puma, Li-Ning, Anta, Decathlon, Mizuno, Yonex, Reebok
	Lesser-known Brands: Xio, Nix, Belock, Shufei, Haosha, Lash, Bart, Senqiong, Diado, Claff
**Sports Activities**	Basketball, Soccer (Football), Badminton, Volleyball, Table Tennis, Mountaineering, Running, Boxing

**Table 2 brainsci-14-00940-t002:** Mean Measurements of Matching between Sports Brands and Sports Activities (Partial).

Sports Brand	Sports Activity	Mean	Sports Brand	Sports Activity	Mean
Nike	Basketball	4.1412	Adidas	Table tennis	2.1655
Adidas	Football	3.3078	Li Ning	Marathon	2.4197
Nike	Football	3.8626	Li Ning	Table tennis	2.6154
Adidas	Basketball	3.8457	Puma	Basketball	2.6949
Li Ning	Volleyball	3.7968	Anta	Basketball	3.6910
Nike	Running	3.7346	Puma	Running	3.9061

**Table 3 brainsci-14-00940-t003:** Matching Degree Between Sports Activities and Sports Brands.

Condition	Sports Activities and Sports Brands
Matching conditions	Nike: Running, Soccer (Football), Basketball, Badminton, Mountaineering Adidas: Soccer (Football), Basketball, Mountaineering Under Armour: Basketball, Badminton, Boxing, Table Tennis Puma: Badminton, Table Tennis Li-Ning: Basketball, Mountaineering Anta: Basketball, Badminton, Mountaineering Decathlon: Mountaineering, Boxing Mizuno: Running, Soccer (Football), Table Tennis Yonex: Badminton, Table Tennis Reebok: Running, Basketball, Mountaineering
Non-matching conditions	Nike: Boxing, Volleyball, Table Tennis Adidas: Running, Badminton, Boxing, Volleyball, Table Tennis Under Armour: Running, Soccer (Football), Mountaineering, Volleyball Puma: Running, Soccer (Football), Basketball, Mountaineering, Boxing, Volleyball Li-Ning: Running, Soccer (Football), Badminton, Boxing, Volleyball, Table Tennis Anta: Running, Soccer (Football), Boxing, Volleyball, Table Tennis Decathlon: Running, Soccer (Football), Basketball, Badminton, Volleyball, Table Tennis Mizuno: Basketball, Badminton, Mountaineering, Boxing, Volleyball Yonex: Running, Soccer (Football), Basketball, Mountaineering, Boxing, Volleyball Reebok: Soccer (Football), Badminton, Boxing, Volleyball, Table Tennis

**Table 4 brainsci-14-00940-t004:** Descriptive Statistics of Purchase Rates across Different Sports Background Groups.

Group Matching		Condition Purchase Rate	Non-Matching Condition Purchase Rate
High Sports Involvement Group	Mean	0.630	0.466
	N	28	28
	SD	0.232	0.245
Low Sports Involvement Group	Mean	0.435	0.425
	N	29	29
	SD	0.254	0.269

**Table 5 brainsci-14-00940-t005:** Descriptive Statistics of Reaction Times across Different Sports Background Groups.

Group		Reaction Time in Matching Condition Reaction	Reaction Time in Non-Matching Condition
High sports involvement group	Mean	619.406	616.358
	N	28	28
	SD	86.577	85.336
Low sports involvement group	Mean	633.927	647.679
	N	29	29
	SD	105.653	103.707

## Data Availability

The data presented in this study are available upon request from the corresponding author due to the fact that the data were derived from human subjects in an EEG experiment.

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
