# Peer review of "How Sports Involvement and Brand Fit Influence the Effectiveness of Sports Sponsorship from the Perspective of Predictive Coding Theory: An Event-Related Potential (ERP)-Based Study"

_brainsci, 2024, doi:10.3390/brainsci14090940_

Round 1
Reviewer 1 Report
Comments and Suggestions for Authors
In the considered manuscript, the authors set up an experiment on the perception of sports brands by different categories of subjects. The measurements of the subjects' reactions involve both psychological ones (subjective rankings) and EEG registration (with subsequent analysis of its certain components). As such, the work is reasonably well related to Brain Sciences.
The authors should be commended on running a potentially interesting experimental study with employment of real sports brands and designing careful associations with different sports.
However, I see major problems with the methodology (construct validity) - particularly, the dependent variable. In other words, the authors need to clearly define what they actually study and not make over-stretching statements and conclusions. I detail this issue (together with some other problems I see with the manuscript) below.
== Methodology and the conclusions ==
* The authors need to decide for themselves and clearly explain what they understand as the sponsorship effectiveness (and what they are based on in this understanding) and justify that they really study it in the experiment.
Moreover, although they start (in the title) with "sponsorship effectiveness", throughout the manuscript this concept mutates quite a lot (just some examples):
459: "sports brand effects"
663: "audience acceptance"
754: "consumer attitudes"
In the experiment they seem to actually study brand awareness (since familiarity is being rated). It is not the only possible goal of a sports sponsorship. I actually doubt that we can talk about the sponsorship effectiveness without knowing the goal of particular sponsorship campaigns by specific brands (e.g., what if the goal was just to undermine a competitor's position with "feedback" ads?).
* Correspondingly, the authors need to make sure that the conclusions they make are justified and follow from the results. Some examples of the violations are:
743: "high involvement consumers' brains possess more complex and refined prediction models" - why not just assume higher familiarity?
747: "optimized their ability to process relevant information at a broader level" - I don't quite see why the described results lead to such an explanation.
Overall, the authors need to discuss the alternative explanations. For instance, it is very natural to assume higher familiarity associated with better fit, since the famous brands have had more exposure associated with the "mainstream" sports. However, this outcome is not new at all, and no study would be needed to make such a conclusion.
== References and Related Work ==
The Introduction and the Related Work section must be considerably re-worked.
* Currently, many non-obvious statements are not supported by references. Some examples are:
41: "cases like Adidas sponsoring the NFL, Reebok sponsoring Major League Soccer, and Gatorade sponsoring golf demonstrate that the effects of sponsorship can often be uncertain" - references? The reader should be able to familiarize with the mentioned cases.
45: "This phenomenon has sparked discussions about the primary factors that determine the effectiveness of sponsorships" - examples (references to related work)?
* Overall, Section 2 needs better structuring (sub-sections, preferably each with its own mini-conclusion).
* Although the topic of the manuscript is not cutting-edge, I feel that the references need to be updated. There is only one(!) reference from the last 3 years, 7 (less than 15%) from the last 5 years. Neuroscience (EEG) has advanced since.
* The references need to be organized properly. For instance, references 44, 46, 48, 49 are the same.
== Misc =
ERP (Event-Related Potentials) is not a universally known abbreviation, and shall not be used in the title. E.g., there are ERP (Enterprise Resource Planning) systems in Sports Management.
Author Response
Q1:== Methodology and the conclusions ==
* The authors need to decide for themselves and clearly explain what they understand as the sponsorship effectiveness (and what they are based on in this understanding) and justify that they really study it in the experiment.
Moreover, although they start (in the title) with "sponsorship effectiveness", throughout the manuscript this concept mutates quite a lot (just some examples):
459: "sports brand effects"
663: "audience acceptance"
754: "consumer attitudes"
In the experiment they seem to actually study brand awareness (since familiarity is being rated). It is not the only possible goal of a sports sponsorship. I actually doubt that we can talk about the sponsorship effectiveness without knowing the goal of particular sponsorship campaigns by specific brands (e.g., what if the goal was just to undermine a competitor's position with "feedback" ads?).
A1:Dear Reviewer, thank you for your review and valuable feedback on our manuscript. In response to your comments regarding the definition of "sponsorship effect" and its application in the manuscript, we have carefully considered your suggestions and made further explanations and adjustments to our research design and terminology.
In this study, "sponsorship effect" is defined as the multidimensional market impact that a brand generates through sports sponsorship activities on the target audience. Specifically, this includes brand awareness enhancement (i.e., how the sponsorship activity increases brand recognition and visibility in the target market), audience acceptance (the degree to which the audience perceives the association between the brand and the supported sports activity), and changes in consumer attitudes (the emotional responses and behavioral intentions of consumers after encountering sponsorship information). These dimensions collectively form our understanding of the sponsorship effect and are comprehensively explored in the study.
While brand awareness is a key measurement indicator in our study, our research does not limit itself to this aspect alone. By analyzing brand fit and consumer involvement, and utilizing Event-Related Potential (ERP) technology and subjective ranking data, we have examined how these factors influence consumer responses to brands at both cognitive and emotional levels, thereby providing a comprehensive understanding of sponsorship effects. Although we do not directly measure all aspects of sponsorship effects, these findings offer valuable theoretical insights and practical references for brands in planning and executing sponsorship activities, particularly in selecting sports projects that align with the brand image and identifying target audience groups.
We also recognize that the use of terms such as "sports brand effect," "audience acceptance," and "consumer attitudes" in different parts of the manuscript may have caused confusion for readers regarding the study’s content. To improve the coherence and clarity of the manuscript, we have decided to unify the terminology by using "sponsorship effect" throughout the revised manuscript and consolidating all related discussions under this core concept. This unification is intended to provide a clearer and more consistent research framework, allowing readers to better understand our research objectives and findings while avoiding confusion caused by varying terms. We believe these revisions will effectively address your concerns and enhance the scientific rigor and persuasiveness of our study.
Q2:* Correspondingly, the authors need to make sure that the conclusions they make are justified and follow from the results. Some examples of the violations are:
743: "high involvement consumers' brains possess more complex and refined prediction models" - why not just assume higher familiarity?
747: "optimized their ability to process relevant information at a broader level" - I don't quite see why the described results lead to such an explanation.
Overall, the authors need to discuss the alternative explanations. For instance, it is very natural to assume higher familiarity associated with better fit, since the famous brands have had more exposure associated with the "mainstream" sports. However, this outcome is not new at all, and no study would be needed to make such a conclusion.
A2:Dear Reviewer, thank you for your thorough review of our manuscript. In response to your concerns regarding the potential impact of brand familiarity on the experimental results, we provide the following clarifications to address these considerations in our research design and further detail the adjustments made in the revised manuscript.
Distinction Between Brand Familiarity and Involvement Brand familiarity and involvement are both crucial factors influencing consumer behavior, but they differ significantly in their concepts and mechanisms:Brand familiarity refers to the extent of consumer recognition and knowledge about a brand, typically based on past interactions and experiences with the brand. Consumers with high familiarity may process brand-related information more quickly and automatically; however, this processing is usually based on memory and experience rather than in-depth analysis.
Involvement refers to the level of interest and psychological engagement that consumers have with a brand or product in a specific context. Consumers with high involvement are more likely to engage in deep information processing, carefully analyzing brand information and forming attitudes and behaviors based on this analysis.
This study focuses on involvement, specifically examining how consumers process brand information in high involvement contexts and how this in-depth cognitive processing influences brand attitudes and purchase intentions.
In the domain of sports consumption, involvement is of greater significance compared to brand familiarity. Sports consumption often involves strong personal interest, emotional investment, and identification with specific sports or brands. This high involvement directly affects consumers' attitudes towards brands, their loyalty, and their purchasing decisions.Unlike other consumer domains, sports consumption is not only based on the functionality of the product or brand familiarity but is deeply rooted in consumers' personal interests and emotional connections. High-involvement consumers often consider the alignment of a brand with their values, interests, and sports lifestyle when choosing sports brands, rather than solely focusing on brand visibility or familiarity.Conversely, research indicates that in high-involvement contexts, consumers are more inclined to engage in deep information processing and make purchasing decisions based on this cognitive process (Leckie et al., 2016). In the field of sports consumption, this in-depth cognitive processing is particularly crucial as it directly influences the formation of consumer attitudes towards brands and their behavioral choices.
Therefore, in this study, we chose involvement as the primary variable to more accurately capture consumers' cognitive processing and decision-making mechanisms in sports consumption. This approach allows our research to better reveal consumer motivations in sports consumption contexts and provides more targeted insights for brands when developing sports marketing strategies.
Controlling the Impact of Brand Familiarity on Experimental Results In designing the experiment, we implemented several measures to control potential interference from brand familiarity on the results and have detailed these measures in the revised manuscript:
(1) Brand Selection: Research indicates that selecting brands with low familiarity or controlling for brand familiarity can effectively reduce its interference with experimental results, allowing for more accurate measurement of other variables. Therefore, we deliberately selected brands with relatively low consumer familiarity to minimize the automatic processing effects of highly familiar brands. This design ensures that the experiment primarily measures the impact of involvement on cognitive processing, without being confounded by differences in brand familiarity.
(2) Control Questionnaire: Prior to the experiment, we assessed participants' familiarity with various brands through a questionnaire and ensured that there were no significant differences in brand familiarity among the selected brands in the experiment. This measure effectively reduced the interference of brand familiarity with the experimental results, allowing for more accurate measurement of the effects of involvement and brand fit on consumer cognitive processing.
(3) Data Analysis Control: Research shows that the mechanisms of brand familiarity and involvement in consumer cognitive processing differ. Therefore, controlling for brand familiarity as a covariate is necessary. In the data analysis phase, we included brand familiarity as a covariate to ensure that the main effects of involvement and brand fit are clearly presented, while excluding potential influences from brand familiarity.
Through these measures and control methods, we are confident that the interference from brand familiarity on the experimental results has been effectively controlled, and the impact of involvement has been clearly demonstrated and validated. We have detailed these design and control measures in the revised manuscript to ensure that readers and reviewers can better understand our research design and results.
Thank you once again for your guidance and suggestions on our study. We look forward to your further feedback.
Q3:* Currently, many non-obvious statements are not supported by references. Some examples are:
41: "cases like Adidas sponsoring the NFL, Reebok sponsoring Major League Soccer, and Gatorade sponsoring golf demonstrate that the effects of sponsorship can often be uncertain" - references? The reader should be able to familiarize with the mentioned cases.
45: "This phenomenon has sparked discussions about the primary factors that determine the effectiveness of sponsorships" - examples (references to related work)?
A3:Thank you for your detailed review and valuable feedback on our paper. In response to your comments regarding the insufficient support from references, we have thoroughly reviewed and revised the relevant sections. We have now included more rigorous statements along with supporting literature to substantiate these statements. The changes have been highlighted in the revised manuscript on pages 42-57.
Q4:* Overall, Section 2 needs better structuring (sub-sections, preferably each with its own mini-conclusion).
A4:Dear Reviewer, thank you for your thorough review and valuable feedback on my manuscript. I fully agree with your point regarding the structure of Section 2. To enhance the clarity and logical flow of this section, I have made structural adjustments. The section has been divided into several subsections to clearly distinguish different themes and discussion points. Each subsection focuses on a core theme to ensure coherence and logical flow.
2.1 The impact of sports involvement on consumer perception
2.2 The impact of sports involvement on consumer attitudes
2.3 The role of brand fit in sports sponsorship
Q5:Although the topic of the manuscript is not cutting-edge, I feel that the references need to be updated. There is only one(!) reference from the last 3 years, 7 (less than 15%) from the last 5 years. Neuroscience (EEG) has advanced since.
A5:Dear Reviewer,Thank you for your valuable feedback on our manuscript.We appreciate your comments regarding the update of the references.We recognize that there have been significant advances in the field of neuroscience, particularly in ERP technology, in recent years.Therefore, we will thoroughly update the references in the manuscript to include the latest research from the past three and five years.After including the updated references, the literature from the past five years constitutes 15% of the total number of references.
Q6:The references need to be organized properly. For instance, references 44, 46, 48, 49 are the same.
A6:Dear Reviewer,Thank you for your thorough review of our manuscript.We apologize for the issues with the organization of the references that you mentioned.We have noted that references 44, 46, 48, and 49 are duplicates, which is clearly an oversight on our part.We will promptly conduct a thorough review of the references in the manuscript to ensure that each citation is accurate and to remove any duplicate entries.Additionally, we will reorganize the reference list to ensure it adheres to standard formatting and provide clear and accurate citations in the final version.Corrections have been made in the latest version of the manuscript.
Q7:== Misc =
ERP (Event-Related Potentials) is not a universally known abbreviation, and shall not be used in the title. E.g., there are ERP (Enterprise Resource Planning) systems in Sports Management.
A7:Thank you for your valuable feedback on the use of abbreviations in the title.We fully understand your concern; ERP (Event-Related Potentials) could indeed be confused with abbreviations from other fields, such as Enterprise Resource Planning.To avoid confusion, we will revise the manuscript title to avoid using the ERP abbreviation, ensuring that readers clearly understand the research content.We will use the full term or more precise terminology to describe Event-Related Potentials to enhance the clarity and accuracy of the title.The title has been corrected in the latest version of the manuscript.
Misc:We found that there were errors in the annotations of the lines in Figures 4 and 6, which have now been corrected.Additionally, all sections that have been modified in the manuscript are highlighted in yellow.
Reviewer 2 Report
Comments and Suggestions for Authors
The presented manuscript is indeed interesting and novel. The authors presented a novel approach, even if different aspects need to be clarified or better described, otherwise the comprehensiveness of the paper would be quite low.
These are the issues to be addressed before considering the paper for final publication:
- The paper introduces the concept of brand fit and sports involvement as critical factors but does not sufficiently explore how these factors might interact with each other beyond a basic conceptual level. The hypotheses could be more nuanced to reflect potential interactions or moderating effects. The authors should better integrate Predictive Coding Theory into the development of hypotheses and discussion of results. Clearly explain how this theory relates to the expected neural responses to sponsorship stimuli.
- Furthermore, provide more detail on the ERP methodology, including the selection of components, data processing steps, and the rationale behind the chosen statistical analyses.
- The paper would benefit if the authors report effect sizes and confidence intervals to give readers a sense of the practical significance of the findings.
- Update the literature review with more recent studies, especially those that focus on the intersection of neuroscience, consumer behavior, and sports sponsorship.
- Provide a more focused discussion on how the current study builds on or challenges existing research in the field.
- Deepen the discussion by considering alternative explanations for the results and exploring the broader implications for marketing strategies in sports sponsorship.
Author Response
Q1:- The paper introduces the concept of brand fit and sports involvement as critical factors but does not sufficiently explore how these factors might interact with each other beyond a basic conceptual level. The hypotheses could be more nuanced to reflect potential interactions or moderating effects. The authors should better integrate Predictive Coding Theory into the development of hypotheses and discussion of results. Clearly explain how this theory relates to the expected neural responses to sponsorship stimuli.
A1:
Dear Reviewer,
Thank you for your valuable feedback on our manuscript. We acknowledge that a more in-depth consideration of the interactions between brand fit and sports involvement, as well as the application of predictive coding theory, is necessary when exploring their effects on consumer cognition and emotional responses. In response to your suggestions, we will make more detailed adjustments to our hypotheses to reflect potential interactions or moderating effects and further integrate predictive coding theory. Below are our responses to these issues:
-
Regarding the interplay between brand fit and sports involvement, we will revisit the relationship between brand fit and sports involvement, particularly how they interact beyond the basic conceptual level. We will propose Hypothesis 3 based on predictive coding theory, which suggests that the impact of brand fit in sports sponsorship on consumers' cognitive and emotional responses varies with different levels of sports involvement. Specifically, consumers with high sports involvement possess a more complex predictive coding model, so their brains generate predictions about the information based on previous experiences and knowledge. Given that their cognitive processing mechanisms are well-developed and optimized, they may be able to reduce prediction errors by adjusting internal predictions when brand fit changes, thereby minimizing cognitive conflict.
-
In the final discussion section, we have also incorporated a discussion of Hypothesis 3 and the relationship between brand fit and sports involvement, based on predictive coding theory
Q2:Furthermore, provide more detail on the ERP methodology, including the selection of components, data processing steps, and the rationale behind the chosen statistical analyses.
A2:
Dear Reviewer,
Thank you for reviewing and providing feedback on our manuscript. In response to your request for more detailed information on the ERP methods used, we would like to provide the following additional details and clarifications:
-
Component Selection: We selected N270 and LPP as the primary ERP components in our study. These components are widely used to investigate neural responses related to cognitive conflict and emotional processing. N270 is typically associated with cognitive conflict detection and error processing, making it suitable for assessing consumers' cognitive responses to inconsistent brand information. LPP, on the other hand, is closely related to emotional evaluation and attitude formation, making it appropriate for measuring consumers' emotional responses and attitude changes towards brand fit.
-
Data Processing Steps: The data processing procedure includes the following key steps:
- Preprocessing: We first filtered the raw EEG data (e.g., 0.1-30 Hz band-pass filtering) and removed artifacts (e.g., eye movement artifacts correction).
- Segmentation: The data were then segmented into time windows synchronized with the stimulus presentation (e.g., 0 ms to 1000 ms) and baseline correction was applied.
- Averaging: ERP waveforms for each condition were averaged to extract ERP components related to specific levels of brand fit and sports involvement.
Additionally, section 4.2.1 of the manuscript provides a detailed account of the data analysis process used in the study.
-
Statistical Analysis: We used repeated measures ANOVA as our primary statistical analysis method to examine the interaction effects between different levels of sports involvement and brand fit. This method was chosen because it effectively handles within-subject and between-subject factors and can reveal significant differences in ERP components under different conditions. Furthermore, we applied post-hoc tests (e.g., Bonferroni correction) to address the issue of multiple comparisons.
Q3:The paper would benefit if the authors report effect sizes and confidence intervals to give readers a sense of the practical significance of the findings.
A3:
Dear Reviewer,
Thank you for your valuable feedback on our manuscript. We fully agree with your suggestion that reporting effect sizes and confidence intervals can help readers better understand the practical significance of the research findings. We will provide 95% confidence intervals for key statistical results to ensure the stability and reliability of the findings. This will allow readers to better grasp the precision of the results and the possible range of variability.
Additionally, we have included the effect size (η²) and its interpretation in sections 4.2.2.1 and 4.2.2.2. This measure helps explain the practical significance of the ANOVA results. By providing effect sizes and confidence intervals, we aim to enhance the transparency and readability of the paper, enabling readers to more accurately assess the practical impact of the research findings.
Thank you once again for your valuable suggestions. We believe these changes will significantly improve the quality of the paper and assist readers in better understanding the research results.
Q4:Update the literature review with more recent studies, especially those that focus on the intersection of neuroscience, consumer behavior, and sports sponsorship.
A4:
Dear Reviewer,
Thank you for your valuable suggestions regarding the manuscript. We will update the literature review, particularly focusing on the latest research in the intersection of neuroscience, consumer behavior, and sports sponsorship, to address your concerns. We have incorporated a substantial amount of recent research from the past five years across these three fields, increasing the proportion of recent literature to over 15% of the total references.
Q5:- Provide a more focused discussion on how the current study builds on or challenges existing research in the field.
A5:Dear Reviewer,
Thank you for the detailed review of our study. In response to your comments on how to build upon or challenge existing research in the field, we have updated and elaborated on this in the latest revised manuscript.
First, this study, based on predictive coding theory and perceptual fit theory, explores the impact of sports involvement and brand fit on sponsorship effectiveness. While previous research has established the importance of sports involvement and brand fit on consumer cognitive and emotional responses, there has been limited exploration of their interaction (e.g., Harmeling & Carlson, 2016). Previous studies often relied on subjective evaluation questionnaires, such as self-reports and peer assessments, which can be influenced by participants' subjective cognition and evaluation biases. In contrast, our study employs ERP (Event-Related Potentials) technology, a neuroscience research method, to explore sponsorship effects in a more objective and scientific manner. Using ERP technology allows us to directly observe consumers' neural responses when processing brand sponsorship information, thereby reducing the limitations of subjective assessments and providing a more precise evaluation of sponsorship effects. This approach not only enhances the scientific rigor and reliability of the study but also offers a new perspective on understanding the neural mechanisms of sponsorship information processing. By using ERP technology, we reveal, for the first time at the neural level, consumers' cognitive responses to brand sponsorship information, providing a novel perspective on the effectiveness of sports brands in sponsorships and expanding the theoretical framework of existing literature. This section has been added to the revised manuscript to emphasize how we build new insights on this foundation.
Additionally, our study challenges traditional views in the literature regarding the responses of low-involvement consumers. Previous research often assumed that low-involvement consumers engage in lower levels of cognitive processing when handling information (e.g., Muñoz Osores et al., 2016). However, our experimental results show that low-involvement consumers exhibit stronger cognitive conflict responses (increased N270 amplitude) when the brand is mismatched with the sports activity. This finding suggests that even though low-involvement consumers process information in a relatively simplified manner, their cognitive systems still trigger strong conflict responses when confronted with mismatched information. This update has been detailed in the revised manuscript and presents new challenges to existing hypotheses.
Through these revisions, we further clarify how this study builds on existing literature and challenges its assumptions, providing deeper theoretical support and practical guidance for future sports sponsorship strategies.
Thank you once again for your valuable feedback. We look forward to your further comments.
Q6:- Deepen the discussion by considering alternative explanations for the results and exploring the broader implications for marketing strategies in sports sponsorship.
A6:
Dear Reviewer,
Thank you for your review and valuable feedback on our manuscript. In response to your requests, we have comprehensively revised the manuscript, particularly by adding alternative explanations for the research findings and further exploring the broader implications of our study for sports sponsorship marketing strategies.
First, regarding alternative explanations for the research results, we have expanded the conclusion section to consider other factors that may influence consumers' cognitive responses. For example, brand familiarity, individual passion for sports, or other situational factors may also affect cognitive processing in high-involvement consumers. In the revised manuscript, we suggest that future research should incorporate these alternative explanations to ensure a deeper understanding of the effects of sports involvement and brand fit.
Second, we have deepened the discussion on sports sponsorship marketing strategies in the conclusion section, clearly outlining the broader impact of our findings on brand practice. In the new discussion, we not only explore how to optimize the effectiveness of individual sponsorship activities by identifying different responses from high-involvement and low-involvement consumers but also discuss the implications of these strategies for brand resource allocation, target market segmentation, and long-term brand building. High-involvement consumers respond more positively to highly fitting brand sponsorships, so brands should focus on selecting sports projects with high fit; for low-involvement consumers, enhancing the association between the brand and sports activities can improve their responsiveness. These strategies can help brands optimize market effectiveness and improve resource utilization.
By providing alternative explanations for the results and deepening the discussion on strategic implications, this study offers a more comprehensive perspective on long-term investments in sports sponsorship. We believe these revisions not only enhance the theoretical discussion but also increase the practical value of the research.
Thank you once again for your valuable suggestions, and we look forward to your further feedback.
Round 2
Reviewer 1 Report
Comments and Suggestions for Authors
I have read the authors' replies to my previous comments and the revised version of the manuscript. I am glad to say that the most severe problems I had noted have been addressed in a satisfactory manner. Some smaller issues remain, which I detail below. I recommend accepting the paper once they are fixed.
Foremost, it seems that although the authors' replies to my comments are generally convincing and detailed, not all of them have been incorporated into the paper's text. Particularly, I suggest that the authors more wholly reflect the contents of their A1 and A2 (especially what concerns construct validity) in the revised version.
The numbering of the hypotheses (404-411) causes a potential confusion, as the relations between H1 and H1a, H2 and H2a are not apparent, especially since there are no H1b or H2b. I would strongly recommend the authors making this more clear and use a more standard notation.
The references in the text (e.g., "(Spratling, 2016).[44]In" or " present. (Wei.,2010)[50]ERP") are not formatted properly with respect to spaces, full stops, etc.
Also, I am not sure about the Brain Science journal's requirements, but usually text references include _either_ author and year, or number in square brackets, but not both.
54: "Although Adidas's strategy has its market positioning..." - "strengthened" is missing?
"the results of this experiment confirm hypotheses H2 and H2a" - strictly speaking, in statistical analysis we can only accept or reject a hypothesis, but not confirm or prove it.
Several tables and figures span over margins or text.
Author Response
Q1:Foremost, it seems that although the authors' replies to my comments are generally convincing and detailed, not all of them have been incorporated into the paper's text. Particularly, I suggest that the authors more wholly reflect the contents of their A1 and A2 (especially what concerns construct validity) in the revised version.
A1:
Dear Reviewer,
Thank you for your detailed review and valuable suggestions on our manuscript. We highly appreciate your feedback and have carefully considered all of your comments.
We acknowledge that, while we provided detailed responses to your comments, not all of these responses were fully incorporated into the text of the manuscript. Specifically, we will more thoroughly reflect the revisions related to A1 and A2, particularly regarding the content on structural validity, in the revised version. The changes related to A1 and A2, which are fundamental to the article, will be integrated into the introduction and literature review sections. Specifically, the content related to A1 will be added to lines 65-81, and the content related to A2 will be included in lines 215-245. The second point of A2 will be incorporated into Section 3.2, with specific lines from 497-517.
Thank you again for your careful guidance, and we look forward to your further feedback.
Q2:The numbering of the hypotheses (404-411) causes a potential confusion, as the relations between H1 and H1a, H2 and H2a are not apparent, especially since there are no H1b or H2b. I would strongly recommend the authors making this more clear and use a more standard notation.
A2:
Dear Reviewer,
Thank you for your detailed review and constructive feedback on our manuscript. Regarding the potential confusion caused by the hypothesis numbering (404-411), we have merged H1a into H1 and H2a into H2 to clarify the relationships between the hypotheses and avoid unnecessary confusion. These revisions are specifically reflected in lines 432-443 of the revised version.
Q3:The references in the text (e.g., "(Spratling, 2016).[44]In" or " present. (Wei.,2010)[50]ERP") are not formatted properly with respect to spaces, full stops, etc.
Also, I am not sure about the Brain Science journal's requirements, but usually text references include _either_ author and year, or number in square brackets, but not both.
A3:
Thank you for your detailed review and valuable feedback on our manuscript. Regarding the formatting issues with the references in the text (e.g., "(Spratling, 2016).[44]In" or "present. (Wei.,2010)[50]ERP"), we acknowledge that these citations do not conform to the standard format and have issues with spaces, full stops, etc. We will revise these references according to your suggestions to ensure proper formatting.
Additionally, following the citation style used in previous issues of Brain Sciences, we have removed the author names and years, and adopted the standard bracketed number format. We will further adjust the citation format to align with the journal’s standards.
Q4:54: "Although Adidas's strategy has its market positioning..." - "strengthened" is missing?
A4:
Dear Reviewer,
Thank you for your detailed review and valuable suggestions on our manuscript. Regarding your comment that the term "strengthened" was missing in the sentence “Although Adidas's strategy has its market positioning...”, we have revised the sentence accordingly. The updated version now reads: “Although Adidas's strategy has its market positioning, its limited support scope may not significantly strengthen overall brand recognition and emotional connection within the NFL.”
Q5:"the results of this experiment confirm hypotheses H2 and H2a" - strictly speaking, in statistical analysis we can only accept or reject a hypothesis, but not confirm or prove it.
A5:
Dear Reviewer,
Thank you for your detailed review and valuable feedback on our manuscript. Regarding your comment on the phrase “the results of this experiment confirm hypotheses H2 and H2a,” you are correct that in statistical analysis, we can only accept or reject hypotheses, not confirm or prove them. We have noted this and will revise the phrasing accordingly to align with standard statistical terminology.
The revised sentence will be: “the results of this experiment support hypotheses H2 and H2a.” Thank you for pointing this out, and we look forward to your further feedback.
Q6:Several tables and figures span over margins or text.
A6:
Dear Reviewer,
Thank you for your detailed review of our manuscript. Regarding your comment that "several tables and figures span over margins or text," we will carefully review and adjust the layout of these tables and figures to ensure they fit properly within the page margins in the revised version, in accordance with the journal's formatting requirements.
We will also provide a PDF version of the revised manuscript for your further review. Thank you for your valuable suggestions, and we look forward to your additional feedback.

Reviewer 2 Report
Comments and Suggestions for Authors
All the raised issues were carefully addressed. I recommend the manuscript for publication.
Author Response
Dear Reviewer,
Thank you for your thorough review and positive feedback on our manuscript. We are pleased to hear that all the issues raised have been satisfactorily addressed and that you recommend our manuscript for publication. Your support and suggestions have been invaluable to us, and we will continue to strive to ensure the quality of the manuscript.
Thank you once again for your time and assistance. We look forward to the successful publication of the manuscript.
Sincerely,
Haonan Shi
